# Progress Overview of Bacterial Two-Component Regulatory Systems as Potential Targets for Antimicrobial Chemotherapy

**DOI:** 10.3390/antibiotics9100635

**Published:** 2020-09-23

**Authors:** Hidetada Hirakawa, Jun Kurushima, Yusuke Hashimoto, Haruyoshi Tomita

**Affiliations:** 1Department of Bacteriology, Graduate School of Medicine, Gunma University, Maebashi Gunma 371-8511, Japan; kurushimaj@gunma-u.ac.jp (J.K.); yhashimoto@gunma-u.ac.jp (Y.H.); tomitaha@gunma-u.ac.jp (H.T.); 2Laboratory of Bacterial Drug Resistance, Graduate School of Medicine, Gunma University, Maebashi Gunma 371-8511, Japan

**Keywords:** two-component regulatory system, virulence, drug resistance, infection control, environmental response and adaptation, histidine kinase, response regulator

## Abstract

Bacteria adapt to changes in their environment using a mechanism known as the two-component regulatory system (TCS) (also called “two-component signal transduction system” or “two-component system”). It comprises a pair of at least two proteins, namely the sensor kinase and the response regulator. The former senses external stimuli while the latter alters the expression profile of bacterial genes for survival and adaptation. Although the first TCS was discovered and characterized in a non-pathogenic laboratory strain of *Escherichia coli*, it has been recognized that all bacteria, including pathogens, use this mechanism. Some TCSs are essential for cell growth and fitness, while others are associated with the induction of virulence and drug resistance/tolerance. Therefore, the TCS is proposed as a potential target for antimicrobial chemotherapy. This concept is based on the inhibition of bacterial growth with the substances acting like conventional antibiotics in some cases. Alternatively, TCS targeting may reduce the burden of bacterial virulence and drug resistance/tolerance, without causing cell death. Therefore, this approach may aid in the development of antimicrobial therapeutic strategies for refractory infections caused by multi-drug resistant (MDR) pathogens. Herein, we review the progress of TCS inhibitors based on natural and synthetic compounds.

## 1. Introduction

The shortage of antimicrobial agents caused by the increasing resistance of bacterial pathogens to current antibiotics is a severe concern in clinical settings. It is estimated that ten million people will die of drug-resistant infections in 2050 if this shortage is not solved [1]. The most commonly used antibiotics were introduced in the twenty years that followed the discovery of penicillin, and their widespread application has resulted in the development of drug-resistant bacteria. Therefore, there is a need to develop new classes of antibiotics, but the process is a challenging one.

The two-component regulatory system (TCS) constitutes a system that enables bacteria to adapt to diverse environmental changes. This system is ubiquitous in bacteria, including animal and plant pathogens, but it is not present in mammalian cells, although some eukaryotic cells, such as fungi, yeast and several higher plant cells, including *Arabidopsis thaliana*, possess similar systems [2,3]. There are many reports that TCSs are associated with bacterial pathogenesis and/or drug resistance/tolerance in addition to fitness in a particular environment [4,5]. Additionally, some TCSs are needed for bacterial cell growth and survival [6,7,8,9,10]. Therefore, these systems have been proposed as potential targets for the development of antibacterial agents.

The TCS essentially comprises two types of proteins, termed “the sensor kinase” (also called “the histidine kinase”) and “the response regulator” [11,12]. In signal transduction, the kinase senses an external stimulus, such as a change in nutrient levels, osmotic pressure, pH, redox state, particular signal molecules, antibiotics or membrane stress. Afterward, a conserved histidine residue in the sensor kinase is phosphorylated, followed by the transfer of the phosphate group to a conserved aspartate residue in the cognate response regulator [11]. Sensor kinases are transmembrane proteins (mostly homodimers) that exhibit auto-phosphorylation and phosphotransferase activities [13]. Consequently, the response regulators demonstrate conformational changes when phosphorylated, then the helix-turn-helix (HTH) domain dimer in the regulator protein binds to a target DNA region adjacent to a promoter. Afterward, the regulator alters the gene expression profiles by modulating the activity of the promoter. Some sensor kinases also exhibit phosphatase activity in the absence of a signal and de-phosphorylate response regulators (the schematic diagram is shown in Figure 1) [14,15]. In 1992, unsaturated fatty acids, such as oleic acid, were reported to attenuate TCS activity [16]. These natural compounds inhibited the auto-phosphorylation of the sensor kinase KinA, which is associated with sporulation induction in *Bacillus subtilis*. The first synthetic inhibitors were reported in 1993 and targeted a TCS of *Pseudomonas aeruginosa* (the following section addresses how these compounds impair TCS activity) [17]. Since the first study, a variety of TCS inhibitors have been described (summarized in Table A1 in the Appendix A, and chemical structures of representative inhibitors are drawn in Figure 2 and Figure 3). These compounds could disrupt the TCS through several mechanisms which include: (1) inhibition of sensor kinase activity, (2) inhibition of response regulator activity, (3) sequestration of signal and (4) inhibition of signal generation. In the following sections, we will discuss the progress made in the development of TCS inhibitors and also address their properties.

## 2. Inhibition of Sensor Kinase Activity

In bacteria, the activity of a sensor kinase is dependent on a particular conserved histidine residue. Alternatively, mammalian kinases rely on serine or threonine, for which they are named as “serine/threonine kinases”. Therefore, sensor kinases are considered as a potential target for antibacterial chemotherapy. Typically, inhibitor candidates are screened in vitro by analyzing the kinase auto-phosphorylation with ATP as a phosphate donor, and/or the phosphorylation of the cognate response regulator incubated with a recombinant sensor kinase protein and ATP [18,19].

The halogenated phenyl-thiazole compounds, 2-(2,3,4-trifluorophenyl)-2,3 dihydrothiazol-3-one and 2-(3-chloro, 4-fluorophenyl)-2,3 dihydrothiazole-3-one, were screened from a synthetic compound library and were the first kinase synthetic inhibitors to be reported. These compounds were shown to inhibit the AlgR2 sensor kinase. This enzyme catalyzes its auto-phosphorylation and the phosphorylation of the cognate response regulator AlgR1, which is involved in alginate gene activation in *P. aeruginosa* [17]. Alginate is one of the exopolysaccharides that contributes to *P. aeruginosa* biofilm formation [20]. The two synthetic inhibitors also inhibited the activity of CheA, NtrB (previously called NRII) and KinA, which are kinases associated with bacterial chemotaxis and nitrogen assimilation in *E. coli*, and sporulation in *B. subtilis*, respectively [21,22,23]. Furthermore, the inhibitors could target the VanS sensor kinase, and reduced its phosphorylation activity towards the cognate response regulator VanR [24]. The latter activates the expression of van genes which contribute to the resistance to glycopeptide antibiotics, such as vancomycin and teicoplanin, in vancomycin-resistant *Enterococci* (VRE) [25]. VanS is activated by glycopeptides [26,27]. Therefore, the two halogenated phenyl-thiazoles also inhibited glycopeptide resistance.

Following earlier studies, some highly hydrophobic compounds, such as salicylanilides, bis-phenols, benzoxazines, benzimidazoles, cyclohexenes and trityls, were proposed as inhibitor candidates [28,29]. However, most of these compounds exhibited poor selectivity and “non-specific” inhibition mechanisms, such as protein aggregation [30]. After these studies on “non-specific” inhibitors, a thienopyridine (TEP) compound was firstly characterized as a “specific” inhibitor relying on ATP competition, and it was shown to impair the activity of several sensor kinases, including HpkA from *Thermotoga maritima*, VicK from *Streptococcus pneumoniae*, and EnvZ from *E. coli*, but it did not inhibit some common mammalian serine/threonine kinases [31]. Additionally, this compound was not toxic to rat myoblasts.

In addition to TEP, radicicol is another putative ATP competitor. Radicicol was originally characterized as an inhibitor of HSP90 [32,33], which is a member of the GHL proteins, together with gyrase B and MutL [34]. This protein has a unique ATP catalytic domain, termed the Bergerat fold [34]. The crystal structures of several sensor kinase proteins, including PhoQ, revealed topological similarities between the Bergerat fold containing an ATP-binding site of the sensor kinases and that of GHL proteins. Therefore, the proteins that possess the Bergerat fold are currently called GHKL proteins (gyrase, HSP90, histidine (sensor) kinase and MutL), thus predicting GHKL inhibitors could target sensor kinases [34]. In the presence of low Mg^2+^ concentrations, PhoQ is auto-phosphorylated and consecutively activates the PhoP response regulator [35]. Although this TCS was originally characterized as a regulatory system responsible for the adaptation to low Mg^2+^ environments, other studies have suggested its critical role in the pathogenesis of *Salmonella*, such as bacterial invasion of epithelial cells and survival within macrophages [36,37]. Among GHKL inhibitors, radicicol was shown to inhibit the auto-kinase activity of PhoQ [38]. Although its effect was relatively moderate, this GHKL inhibitor could be beneficial for the development of more potent sensor kinase inhibitors. PhoQ also activates the response regulator PmrA via a small connector protein, contributing to the induction of resistance to cationic antimicrobial peptides, such as polymyxin B [39,40]. This antibiotic is often employed as an alternative “last-resort” drug in the treatment of infections caused by Gram-negative bacteria that are resistant to commonly used antibiotics, such as quinolones, aminoglycosides and β-lactams, including carbapenems [41]. Therefore, the inhibitor of PhoQ could be a potential supportive drug in polymyxin B therapy. The PhoP/PhoQ system is vital for the virulence of *Shigella* species, thus making it a possibly efficient target for the treatment of infections caused by these bacteria [42]. Cai et al. identified four inhibitor candidates through a virtual screening based on the putative 3D structure of the ATP catalytic domain of PhoQ from *Shigella flexneri* [43]. These compounds could directly bind to the recombinant cytoplasmic domain of PhoQ and inhibited its kinase activity. Furthermore, these inhibitors suppressed the invasion of HeLa cells by *S. flexneri* without exhibiting apparent cytotoxic effects and hemolytic activities. Additionally, mice administrated with each inhibitor showed no symptoms of inflammation in a Sereny test.

The QseC sensor kinase and the QseB response regulator pair are highly conserved in some Gram-negative pathogens, including enterohaemorrhagic *E. coli* (EHEC), uropathogenic *E. coli* and *Salmonella enterica*. This TCS has been shown to be associated with bacterial pathogenesis [44,45,46]. QseC is auto-phosphorylated in the presence of epinephrine [44]. A benzenesulfonamide compound, named LED209, was screened from a library of small organic molecules and was found to antagonize the auto-phosphorylation of QseC in the presence of epinephrine [47,48]. This compound also decreased the QseB-induced expression of LEE genes which encode a subset of type III secretory proteins required for EHEC pathogenicity and resulted in the attenuation of attaching and effacing (A/E) lesions. In addition to LEE genes, LED209 also suppressed the QseB-induced expression of *fliC* and *stx2* genes, which encode flagellin and Shiga-toxin type 2, respectively [47]. Flagellin is necessary for bacterial motility, contributing to an initial attachment of bacteria to epithelial cells and bacterial fitness in the host while Shiga-toxin type 2 produced by EHEC is closely associated with the induction of hemolytic-uremic syndrome (HUS) [49,50]. Finally, LED209 could also decrease bacterial distribution and mortality of mice infected with *S. enterica* [47].

The TCS comprising WalK sensor kinase and WalR response regulator (previously named YycG and YycF, respectively) is essential for cell growth of some Gram-positive bacteria, such as *S. aureus*, *Enterococcus faecalis* and *B. subtilis* [6,7,9,10]. This TCS contributes to cell wall biosynthesis. Some compounds have been reported as potential inhibitors of WalK. For example, a zerumbone ring-opening compound and an imidazole derivative named NH125 were shown to inhibit the activity of WalK [51,52,53]. However, these compounds are likely “non-specific” inhibitors because these compounds bind to sensor proteins at the four-helix bundle that contains the conserved histidine residue. Consequently, this binding event induces conformational changes that lead to protein aggregation. After these earlier studies, several “specific” WalK inhibitors were reported. For example, di-anthracenone walkmycin B was isolated from methanol extracts of a soil microbe and it was shown to specifically bind to WalK in *B. subtilis* and *S. aureus*, thereby inhibiting the sensor kinase auto-phosphorylation [54]. This compound could inhibit the growth of some multi-drug resistant Gram-positive pathogens, including methicillin-resistant *S. aureus* (MRSA) and VRE. In addition to WalK, walkmycin C (walkmycin B analogue) could inhibit the auto-phosphorylation of CiaH, and LiaS sensor kinases in *Streptococcus mutans* [55]. These sensor kinases are associated with biofilm formation, acid tolerance and the development of genetic competence. These processes were attenuated in *S. mutans* when grown with walkmycin C of sub-MIC levels (1/5 or 1/10 of MIC), although this bacterium still grew normally. Along with traditional in vitro screening from natural sources, several virtual screenings using structural models of WalK have proposed alternative candidates for sensor kinase inhibition. Benzamide, furan, pyrimidinone and imidazole derivatives were identified in screening studies based on the ATP catalytic domain of WalK orthologues (named VicK) from *Staphylococcus epidermidis* and *S. pneumoniae* [56,57]. These compounds inhibited the auto-phosphorylation of VicK and cell growth in *S. epidermidis* and *S. pneumoniae*. Recently, Velikova et al. proposed extensive inhibitor seeds by combining a WalK structure-based virtual screen with an in vitro fragment-based screen via differential scanning fluorimetry and developed nine WalK inhibitors [58]. These compounds could inhibit kinase auto-phosphorylation and exhibited an antibacterial effect against clinical isolates of *S. aureus*, including multi-drug resistant strains. Additionally, signermycin, a decalin ring molecule containing a tetramic acid moiety, and 1-dodecyl-2-isopropylimidazole were proposed as WalK dimerization inhibitors, different from the action mode of walkmycin B [59,60]. These compounds targeted the dimerization site distant from the catalytic ATP binding site, which impaired the dimerization of the WalK protein without causing protein aggregation.

Many Gram-positive bacteria produce short peptides, termed “auto-inducing peptides (AIP)” or “auto-inducers”, that function as signaling molecules and activate a subset of genes, including those which contribute to virulence and genetic competence via particular TCSs [61]. The AIP analogues have been designed as putative kinase antagonists. *S. aureus* strains are divided into four different groups which produce different AIPs. Heterologous AIPs do not activate the AgrC sensor kinase but impair the activation by competing with the autologous AIP [62]. Lyon et al. designed an AIP analogue that antagonized all groups of *S. aureus* AIPs [63]. This peptide was shown to decrease the production of δ-toxin induced by the AIPs. In several studies, synthetic AIP analogues of *S. pneumoniae* were shown to inhibit development of genetic competence and virulence followed by activation of the ComD sensor kinase [64,65,66]. AIP analogues of some other Gram-positive pathogens, such as *S. epidermidis* and *E. faecalis*, were also reported [67,68]. An overview of AIPs and their antagonists was recently reviewed [69].

Novel drugs for the treatment of infections caused by *Mycobacterium tuberculosis* are strongly desired because of the increasing resistance of this species to many kinds of conventional antibiotics (commonly called “multi-drug resistant tuberculosis: MDR-TB” and “extensively drug resistant tuberculosis: XDR-TB”). Some TCSs of *M. tuberculosis* have been proposed to be potential drug targets [70]. One such example is the DevS sensor kinase and the cognate response regulator DevR, which are required for bacterial survival under oxygen-limited conditions associated with periods of non-replicating bacterial dormancy within host cells. Kaur and colleagues found that peptides mimicking the N-terminal receiver domain of DevR reduced bacterial survival in anaerobic cultures [71]. The receiver domain contains the conserved aspartate residue and threonine and lysine residues that are implicated in phosphorylation and conformational changes, respectively. The mimetic peptides were proposed to bind near the DevS phosphorylation site. These peptides, indeed, impaired the auto-phosphorylation of the DevS protein. The NarX/NarL TCS could be another possible drug target in *M. tuberculosis*, for which a small inhibitor candidate, presumably targeting the NarL response regulator, was reported [72]. This compound and its properties are addressed in the following section.

## 3. Inhibition of Response Regulator Activity

The response regulator is an alternative target for the disruption of the TCS. The key events participated in by the response regulator are the phosphorylation of its conserved aspartate residue mediated by the cognate sensor kinase, and the subsequent binding to the target DNA [14,15]. Therefore, the considerable modes of action of the inhibitors are broadly classified into two groups: inhibition of phosphorylation and inhibition of DNA binding. In some cases, the response regulator may be a more efficient target than the sensor kinase because some response regulators can be phosphorylated by “non-cognate” alternative kinases and small-molecule phosphate donors, such as acetyl phosphate, also contribute to the phosphorylation of the response regulator [73,74].

Lactoferricin B was originally characterized as a cationic antimicrobial peptide [75]. In a proteomics study with *E. coli*, two response regulator proteins, BasR and CreB, were identified as lactoferricin B-binding proteins [76]. Pre-incubation of BasR and CreB with the antimicrobial peptide led to the inhibition of the binding to their respective recombinant phosphorylated BasS and CreC sensor kinases. This finding suggested that lactoferricin B targeted BasR and CreB and eventually impaired their phosphorylation. On the other hand, this peptide did not affect the binding of the response regulators to the target DNA. BasR contributes to bacterial survival in the presence of a high concentration of ferric ions while CreB induces the expression of some genes involved in primary metabolism when *E. coli* is cultivated in minimal media [77,78]. In both of these conditions, lactoferricin B was shown to impair the growth of *E. coli* [76].

In silico analysis based on the 3D-structure of the response regulator is an alternative strategy for inhibitor screening. For example, an in silico analysis using the structure of the PhoP response regulator and a chemical library suggested that an anthraquinone compound (named Rhein) could be a potential PhoP inhibitor. This simulation assay indicated that Rhein could bind to the N-terminal receiver domain containing the conserved phosphorylation site, which would block the phosphorylation of PhoP [79]. This anthraquinone compound was originally isolated from several plant species, such as *Rheum undulatum*, and was shown to have antibacterial activity [80]. Thus, Rhein may have another application as a potential TCS inhibitor. Another example of an in silico approach was the molecular docking analysis of the phosphorylation site of NarL from *M. tuberculosis* [72]. Using this strategy, the compound 1-{1-[(3-nitrophenyl) methyl] piperidin-2-yl} ethan-1-amine was predicted to block the phosphorylation of NarL. However, the mentioned in silico studies lacked experimental validation, such as testing if the proposed molecules bound to the target response regulator proteins (PhoP and NarL, respectively) and if the expression of target genes was suppressed in bacteria cultured with the putative inhibitors.

In addition to phosphorylation inhibitors, several compounds that impair the DNA binding activity of response regulators have been reported. Initially, two alkyl imidazole derivatives were isolated [17]. These compounds demonstrated inhibition of the binding of the *P. aeruginosa* AlgR1 response regulator to the target DNA containing the promoter of *algD*, the alginate structural gene, at a concentration of 50 μg/mL, without affecting the phosphorylation of both AlgR1 and the cognate sensor kinase AlgR2. These compounds were suggested to suppress the transcription of *algD* without inhibiting bacterial growth.

Along with WalK inhibitors, inhibitors of the WalR response regulator were also explored. Gotoh et al. found two novel WalR inhibitors: 4-methoxy-1-naphtol and phenyl-pyrimido derivatives, named walrycin A and walrycin B, respectively [81]. This group developed a high-throughput screening system to evaluate the dimerization of WalR and its binding to target DNA. These identified compounds could alter the conformation of the WalR homodimer and reduce its binding affinity to target DNA. Similar to the WalK inhibitors, such as walkmycin B, the two walrycins exhibited antibacterial activity against *B. subtilis* and *S. aureus*.

Recently, tiratricol, propidium iodide, lithocholic acid (LCA) and lorglumide sodium salt were identified as potential inhibitors of the ArsR response regulator from *Helicobacter pylori* by screening the Prestwick Chemical Library, which comprises more than 1100 small molecules [82]. ArsR is phosphorylated by the ArsS sensor kinase under acidic conditions, such as those in the stomach, and it is essential for the growth of *H. pylori* [83,84]. These compounds were shown to inhibit the DNA binding activity of ArsR according to the electrophoretic mobility shift assay (EMSA) and exhibited moderate bactericidal activities. Among these inhibitors, LCA elevated the antibacterial activity of clarithromycin and levofloxacin through additive interactions. *H. pylori* has another response regulator, named HsrA, which is also essential for bacterial viability, although its cognate sensor kinase remains unknown [85]. An analysis of the Prestwick Chemical Library identified seven natural flavonoids as potential inhibitors of HsrA [85]. The compounds interacted with particular amino acid residues in the response regulator which were associated with the formation of the helix-turn-helix DNA binding motif and, consequently, inhibited the DNA binding activity. In addition, some of the flavonoid compounds exhibited strong bactericidal activity against *H. pylori*.

In another example of in silico screening, several aromatic compounds from a drug-like library of the National Cancer Institute in USA were proposed to target the PhoP response regulator [86]. These screened compounds bound to the conserved α4-β5-α5 structural motif of the N-terminal receiver domain that is important for the homo-dimerization and function of PhoP. However, the dimerization and phosphorylation of the response regulator were not affected by binding of the inhibitors. Biochemical analyses, including size-exclusion chromatography (SEC), Forster resonance energy transfer (FRET) analyses and in vitro phosphorylation and electrophoretic mobility shift assay (EMSA), revealed that the aromatic compounds altered the conformation of PhoP when bound to the α5-helix and inhibited the formation of the PhoP–DNA complex [86].

The response regulator interacts with the catalytic domain center of the sensor kinase. Therefore, peptides that mimic the kinase active site may bind to the response regulator and interfere with its function. Following this rationale, Ulijasz et al. used phage display technology to screen a combinational peptide library for ligands which bind to the VanR response regulator. After the screening, the dodecamer peptide E12 (SLCHDSVIGWEC) was selected, and its peptide analogue (SLAHDSIIGYLS), named E12.1, was engineered with a VanS catalytic domain center sequence based on E12 [87]. This peptide could bind to the N-terminus of phosphorylated VanR and inhibit its interaction with the promoter region of the target DNA. Therefore, this study showed that a design of synthetic peptides which targets the response regulator based on the amino acid sequence of the cognate sensor kinase could be an effective strategy in the development of novel TCS inhibitors.

## 4. Other Considerable Mechanisms; Sequestration of Signal and Inhibition of Signal Generation

As an alternative technique for the interference of TCS activity, signal sequestering has been proposed. This concept is based on blocking the activation of sensor kinase in extracellular space without targeting the protein itself. AST-120 is an oral carbonaceous adsorbent that is used in the treatment of progressive chronic kidney disease (CKD). This drug adsorbs indole, which is a precursor of the uremic toxin, produced by enteric bacteria, and alleviates the accumulation of uremic toxin in the kidneys of CKD patients [88]. The CpxA and BaeS sensor kinases sense indole and activate their cognate response regulators, CpxR and BaeR. This leads to an increase in the antibiotic tolerance in *E. coli*, including EHEC, and the production of proteins from the type III secretion system, which are required for virulence of EHEC [89,90,91]. AST-120 was shown to abolish the effect of indole on the induction of drug tolerance and virulence by adsorbing the indole signal [92]. This study suggested a potential application of AST-120 in antibacterial chemotherapy.

The inhibition of signal generation is another strategy for the disruption of the TCS. Herein, we present two examples. The first study described several small molecules, named “COM-blockers”, that blocked competence in *S. pneumoniae*. This species is capable of natural genetic transformation, which is a mechanism that enables the uptake of extracellular DNA and integrates it into the bacterial genome [93,94]. Competence activation in *S. pneumoniae* is regulated by the ComD sensor kinase, the ComE response regulator and its autocrine signal peptide, named the “Competence stimulating peptide” (CSP) [95]. The CSP is secreted via a particular transporter complex driven by the proton motive force and then processed to an active shorter peptide. Domenech et al. identified COM-blockers that disrupted the proton motive force, thereby reducing CSP secretion in *S. pneumoniae*, and resulted in the reduced level of the active short CSP [96]. Although disruption of the proton motive force by the Com-blockers was predicted to potentially cause bacterial growth defect, the COM-blockers were able to inhibit horizontal gene transfer at concentrations that did not affect bacterial viability in an in vivo murine model of infection. The second study focused on 5′-methylthio-(MT-), 5′-ethylthio-(EtT-) and 5′-butylthio-(BuT-) DADMe-ImmucillinAs which are transition state analogues of 5′-methylthioadenosine nucleosidase (MTAN). This MTAN enzyme catalyzes the deadenylation of 5′-methylthioadenosine (MTA) and S-adenosyl homocysteine (SAH), consequently producing 5′-methylthioribose (MTR) and S-ribosylhomocysteine (SRH). The SRH becomes a precursor of autoinducer-2 (AI-2) generation [97]. The DADMe-ImmucillinA compounds have been principally described as inhibitors of AI-2 synthesis [98]. AI-2 is one of the signaling molecules used in bacterial cell-to-cell communication termed “Quorum Sensing”. The AI-2 mediated regulatory mechanism has been well characterized in *Vibrio cholerae*. AI-2 and its accessary receptor protein, LuxQ, dephosphorylate the LuxQ sensor kinase and reduce the level of phosphorylated LuxO response regulator. This leads to the de-repression of ToxT which is repressed by non-coding regulatory RNAs. ToxT is a positive regulator of virulence proteins, including cholera toxins. A 3D-structural analysis of MTAN from *V. cholerae* with BuT-DADMe-ImmucillinA indicated that the inhibitor bound to the catalytic active site of the protein, producing hydrophobic stacking interactions. Finally, these DADMe-ImmucillinA compounds have been shown to inhibit MTAN activity and to reduce AI-2 production in *V. cholerae*.

## 5. Concluding Remarks and Future Prospects

Blocking bacterial TCS is a promising strategy in antibacterial therapy. To date, a variety of compounds have been discovered from natural and synthetic libraries. These agents disrupt the TCS by means which include inhibiting kinase activity, phosphorylation of the response regulator and the action of binding to target DNAs, sequestering signals and impairing signal generation. Some TCS inhibitors function as conventional bactericidal agents by disrupting the activity of essential TCSs, such as WalK/WalR and ArsS/ArsR. These compounds offer a great benefit because these targets and structures are definitely different from those of currently used antibiotics, hence these TCS inhibitors are not affected by the development of cross resistance in currently known MDR pathogens. For instance, WalK and WalR inhibitors have been shown to be effective against MRSA and VRE. On the other hand, inhibitors which suppress bacterial virulence and reduce drug resistance/tolerance rather than killing bacteria have been reported. These inhibitors may expand chemotherapeutic strategies to combat MDR pathogens by a combination with conventional bactericidal therapy. The majority of screening studies to identify new inhibitor candidates have been conducted in vitro by relying on a particular reporter system or high-throughput phosphorylation assays and by using chemical libraries. In addition, recently developed in silico screening strategies have been successful in the identification of putative inhibitors. However, since most of the proposed candidates were only characterized in vitro and ex vivo, their activities need to be validated in animal models of infection. Moreover, the majority of identified inhibitors contained highly hydrophobic structures, such as aromatic rings and/or heterocyclic structures. Therefore, it is crucial to develop processes that would increase the hydrophilicity and solubility of these compounds. Finally, other questions remain unanswered, such as whether these agents only target the TCS, and whether they have any critical and unexpected side effects in addition to their pharmacokinetics (ADME: absorption, distribution, metabolism and excretion). These questions need to be addressed in the near future to enable us to use these agents as novel antibacterial agents.

## Figures and Tables

**Figure 1 antibiotics-09-00635-f001:**
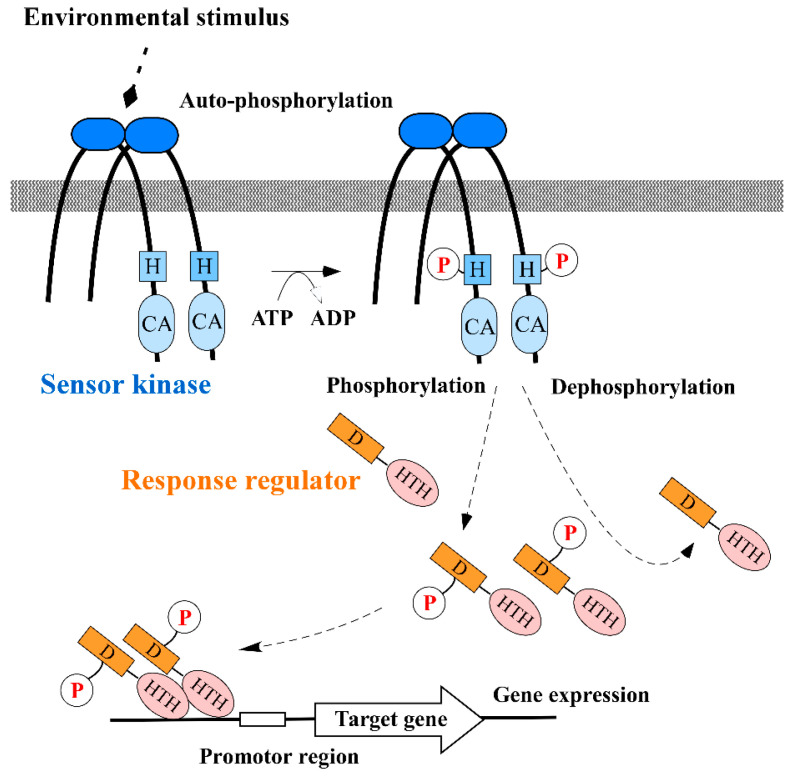
Two-component regulatory system (TCS) in bacteria. When the membrane-bound homo-dimeric sensor kinase senses a particular environmental stimulus, the conserved histidine residue (H) in the cytoplasmic sensor domain of this protein is phosphorylated (P) (termed auto-phosphorylation), then transfers its phosphate group to the conserved aspartate residue (D) in the receiver domain of the cognate response regulator. The kinase activity depends on ATP which is bound to the catalytic domain (CA). The phosphorylated response regulator forms a homo-dimer, then the helix-turn-helix domain (HTH) of the response regulator binds to particular DNA sequences on or close to the promoter of target genes.

**Figure 2 antibiotics-09-00635-f002:**
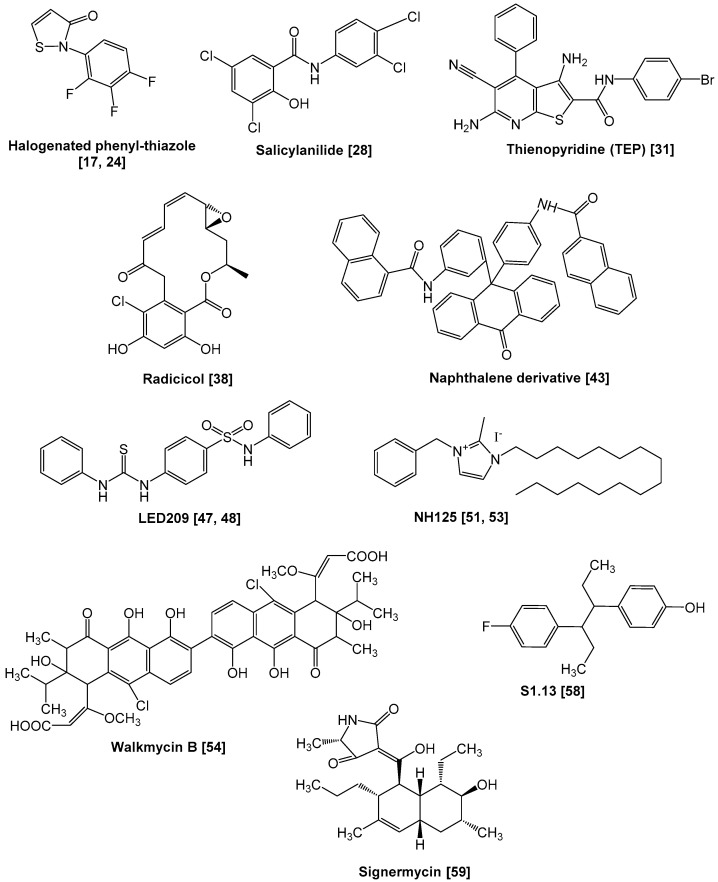
Chemical structures of representative sensor kinase inhibitors.

**Figure 3 antibiotics-09-00635-f003:**
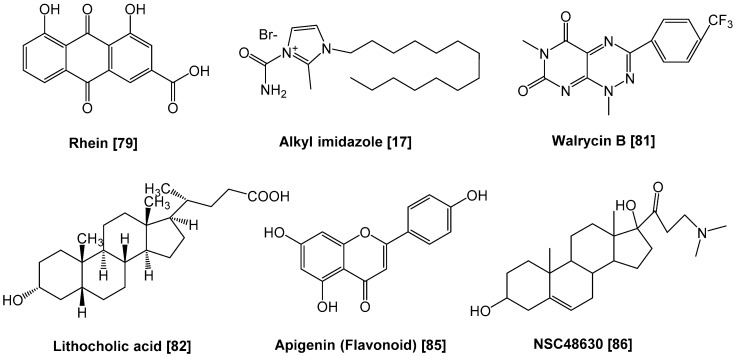
Chemical structures of representative response regulator inhibitors.

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
