# Peer review of "Progress Overview of Bacterial Two-Component Regulatory Systems as Potential Targets for Antimicrobial Chemotherapy"

_antibiotics, 2020, doi:10.3390/antibiotics9100635_

Round 1

Reviewer 1 Report

The review by Hirakawa et al provides a progress overview of the bacterial two-component system as a potential strategy for antimicrobial chemotherapy. The authors concisely report on the inhibition of the sensor kinase and response regulator activities as well as include a small section on other considerable mechanisms. Some changes to the current submission should be considered before publication as stated below.

Major comments:

  1. Section 2 and section 3 - tables that summarise the reported compounds, their targets and references should be included for the benefit of readers. Where possible, schemes can also be included. 
  2. Another section that provides comparison on the reported antimicrobial efficacies of the compounds may be helpful to give a better idea about compounds with greater efficacies or to provide an idea of which approach has shown greater efficacy.
  3. On page 6, the paragraph on the auto-inducer peptides should be expanded if possible as there are more reports on using these compounds in the literature. Currently, the authors state only one paper on these. 
  4. Section 5 on the future prospects of the TCS is rather short, it will be useful if the authors provided a more detailed view or explanation on the future application of these systems as antimicrobial chemotherapeutics and why investigating these will be helpful. 

Minor comments:

  1. Lane 305: "mechanism that enables to the.." should be changes to "mechanism that enables the uptake".
  2. Lane 250 to 252: Sentence should be rephrased as it does not sound grammatically correct. 

Author Response

The review by Hirakawa et al provides a progress overview of the bacterial two-component system as a potential strategy for antimicrobial chemotherapy. The authors concisely report on the inhibition of the sensor kinase and response regulator activities as well as include a small section on other considerable mechanisms. Some changes to the current submission should be considered before publication as stated below.

Response: Thank you for your suggestions and comments. We revised our manuscript according to your suggestions, and added a table to summarize properties of inhibitors.

Major comments:

Section 2 and section 3 - tables that summarise the reported compounds, their targets and references should be included for the benefit of readers. Where possible, schemes can also be included. 

Response: We provided a table that summarize the inhibitors as Table1.

Another section that provides comparison on the reported antimicrobial efficacies of the compounds may be helpful to give a better idea about compounds with greater efficacies or to provide an idea of which approach has shown greater efficacy.

Response: Thank you for your suggestion, a comparison on efficacies among inhibitors could be good idea as an option. However, it is difficult to objectively compare their efficacies because system and detection limit to assay inhibitors are technically different among studies.

On page 6, the paragraph on the auto-inducer peptides should be expanded if possible as there are more reports on using these compounds in the literature. Currently, the authors state only one paper on these. 

Response: As suggested, we added literature related to inhibitors of

AIPs from S. pneumoniae, S. epidermidis and E. faecalis (Ref. 64-68). In

addition, we found that another group recently reviewed AIP studies

including inhibitors (Ref. 69). We now also cited it.

Section 5 on the future prospects of the TCS is rather short, it will be useful if the authors provided a more detailed view or explanation on the future application of these systems as antimicrobial chemotherapeutics and why investigating these will be helpful.

 Response: As suggested, we provided these statements (Please see this section).

Minor comments:

Lane 305: "mechanism that enables to the.." should be changes to "mechanism that enables the uptake".

Response: Corrected.

Lane 250 to 252: Sentence should be rephrased as it does not sound grammatically correct. 

Response: We changed the sentence to This group developed a high-throughput screening system that enables to evaluate abilities of WalR for dimerization and binding to target DNA”.

Reviewer 2 Report

This review gives a succinct account on prior attempts to design inhibitors of two-component systems. The appended manuscript contains some corrections  and some minor comments.

Major comments

I would choose a better title: Progress Overview of Bacterial Two-Component Regulatory Systems as Potential Targets for Antimicrobial Chemotherapy 

line14: Many TCS consist of more than two proteins (WalRK and YycH and YycI, VraSR and VraT....), this fact should be incorporated.

line 18: All bacteria have TCS systems, not only pathogens.

line 51: Please do not omit that many kinases also dephosphorylate their response regulators.

line 100: I personally would appreciate some more information about the substances that lead to aggregation of the kinase in vitro. There is such a confusion in the literature. Is it possible to enumerate these substances or show their structures to obtain a clear picture? This would be an overview that would be very useful and would give more impact to this review. (E. g. closantel was rediscovered as a kinase inhibitor).

line 109: Please check role of vicK in Enterococcus since formerly walK in S. aureus was called vicK. So the compound should have had an effect on growth, please check!

line 115 ff: The Bergerat fold is the ATP-binding site and the proteins that possess it are now called GHKL proteins (gyrase, HSP90, histidine kinase, and MutL)

line 164: The exact binding site is not specified in the paper.

line 186 ff: Here you should perhaps mention that even in nature these AIP antagonists exist: There are 4 different groups of S. aureus which produce different AIPs. Foreign (heterologous) AIPs do not activate the system but bind and inhibit the system of foreign strains.

line 202: How can phosphotransfer function if the step before, autophosphorylation, is inhibited?

line 310: Disruption of proton motive force should have far-reaching consequences for bacteria. Normally these compounds are antibiotic or biocides. Please explain better.

Author Response

This review gives a succinct account on prior attempts to design inhibitors of two-component systems. The appended manuscript contains some corrections  and some minor comments.

Response: Thank you for your suggestions and comments. We revised our manuscript according to your suggestions and attached PDF file.

Major comments

I would choose a better title: Progress Overview of Bacterial Two-Component Regulatory Systems as Potential Targets for Antimicrobial Chemotherapy 

Response: We changed the title as suggested.

line14: Many TCS consist of more than two proteins (WalRK and YycH and YycI, VraSR and VraT....), this fact should be incorporated.

Response: As suggested, we added the word of “at least”.

line 18: All bacteria have TCS systems, not only pathogens.

Response: As suggested, we modified it.

line 51: Please do not omit that many kinases also dephosphorylate their response regulators.

Response: As suggested, we added this statement into the text and Fig1.

line 100: I personally would appreciate some more information about the substances that lead to aggregation of the kinase in vitro. There is such a confusion in the literature. Is it possible to enumerate these substances or show their structures to obtain a clear picture? This would be an overview that would be very useful and would give more impact to this review. (E. g. closantel was rediscovered as a kinase inhibitor).

Response: We provided additional information of inhibitors that may lead to protein aggregation into the text (there are benzoxazines and benzimidazoles and cyclohexenes and trityls in addition to bis-phenol and salicylanides). As far as we understand, Closantel is one of salicylanide derivatives. We provided a chemical structure of salicylanide into Fig.2.

line 109: Please check role of vicK in Enterococcus since formerly walK in S. aureus was called vicK. So the compound should have had an effect on growth, please check!

Response: Ref31 described “TEP did not inhibit a panel of gram-positive and –negative bacteria, including E. faecalis SP409. However, we definitely agree your suggestion. E. faecalis VicK is very important for cell growth. The SP409 strain described in the reference may have an additional TCS compensate VicK??? To avoid the confusion and misleading, we deleted the sentence.

line 115 ff: The Bergerat fold is the ATP-binding site and the proteins that possess it are now called GHKL proteins (gyrase, HSP90, histidine kinase, and MutL)

Response: As suggested, we modified it.

line 164: The exact binding site is not specified in the paper.

Response: As suggested, we modified it.

line 186 ff: Here you should perhaps mention that even in nature these AIP antagonists exist: There are 4 different groups of S. aureus which produce different AIPs. Foreign (heterologous) AIPs do not activate the system but bind and inhibit the system of foreign strains.

Response: As suggested, we added the literature.

line 202: How can phosphotransfer function if the step before, autophosphorylation, is inhibited?

Response: We intended that the inhibitor suppressed auto-phosphorylation of the sensor kinase, but not phosphorylation of the response regulator. To avoid the misleading, we modified the text as that.

line 310: Disruption of proton motive force should have far-reaching consequences for bacteria. Normally these compounds are antibiotic or biocides. Please explain better.

Response: As you suggest, disruption of PMF leads to far-reaching consequences including bacteriostatic or bactericidal effect. However, according to the reference, the authors used inhibitors at concentrations, that did not affect bacterial viability. We added this statement into the text.

Reviewer 3 Report

the review manuscript entitled "Progress Overview of Bacterial Two-Component
Regulatory System as a Potential Target for Antimicrobial Chemotherapy" is a very interesting overview in the antibiotics topic.

i think that this review is well written and organized, with a very intense citation of literature and argumentation of the ihibitors of bacterial growth. The typical readership could benefit of this review in the literature field, facilitating the development of new antibacterial/chemotherapeutic drugs.

In my opinion, english should be revised and images could be of improved quality before publication.

The figure reporting the chemical structures is not convincing; please use a more high place and separate figures. Also, do not use the name of authors but reference number to identify the structures.

Author Response

The review manuscript entitled "Progress Overview of Bacterial Two-Component
Regulatory System as a Potential Target for Antimicrobial Chemotherapy" is a very interesting overview in the antibiotics topic.

I think that this review is well written and organized, with a very intense citation of literature and argumentation of the ihibitors of bacterial growth. The typical readership could benefit of this review in the literature field, facilitating the development of new antibacterial/chemotherapeutic drugs.

In my opinion, english should be revised and images could be of improved quality before publication.

The figure reporting the chemical structures is not convincing; please use a more high place and separate figures. Also, do not use the name of authors but reference number to identify the structures.

Response: Thank you for your suggestions and comments. We fixed our English in some parts, and Fig2 were now separated into two figures to obtain a higher resolution as suggested.

Round 2

Reviewer 1 Report

The authors made significant changes to the manuscript.

I would request the acceptance of the manuscript after a minor grammatical correction. 

Minor correction:

Change sentence in lane 263 from "This group developed a high-throughput screening system that enables to evaluate abilities of WalR for dimerization and binding to target DNA" to

"This group developed a high-throughput screening system to evaluate the dimerization of WalR and its binding to target DNA".